# Effect of Feeding Insoluble Fiber on the Microbiota and Metabolites of the Caecum and Feces of Rabbits Recovering from Epizootic Rabbit Enteropathy Relative to Non-Infected Rabbits

**DOI:** 10.3390/pathogens11050571

**Published:** 2022-05-12

**Authors:** Xiao-Haitzi Daniel Puón-Peláez, Neil Ross McEwan, Roberto Carlos Álvarez-Martínez, Gerardo Mariscal-Landín, Gerardo Manuel Nava-Morales, Juan Mosqueda, Andrea Margarita Olvera-Ramírez

**Affiliations:** 1Doctorado en Ciencias Biológicas, Facultad de Ciencias Naturales, Universidad Autónoma de Querétaro, Avenida de las Ciencias S/N, Juriquilla, Delegación Santa Rosa Jáuregui, Querétaro C.P. 76230, Mexico; xh.puon@gmail.com; 2School of Pharmacy & Life Sciences, Robert Gordon University, Garthdee House, Garthdee Rd, Aberdeen AB10 7GJ, UK; n.mcewan@rgu.ac.uk; 3Facultad de Ciencias Naturales, Universidad Autónoma de Querétaro, Av. Junipero Serra, Antiguo Aeropuerto, Campus Aeropuerto S/N, Santiago de Querétaro, Querétaro C.P. 76140, Mexico; roberto.alvarez@uaq.mx; 4Centro Nacional de Investigación en Fisiología Animal INIFAP, Km. 1, Carretera a Colón, Col. Ajuchitlán Colón, Querétaro C.P. 76280, Mexico; mariscal.gerardo@inifap.gob.mx; 5Facultad de Química, Universidad Autónoma de Querétaro, Cerro de las Campanas S/N, Santiago de Querétaro, Querétaro C.P. 76010, Mexico; gerardomnava@gmail.com; 6Cuerpo Académico Salud Animal y Microbiología Ambiental, Facultad de Ciencias Naturales, Universidad Autónoma de Querétaro, Avenida de las Ciencias S/N, Juriquilla, Delegación Santa Rosa Jáuregui, Querétaro C.P. 76230, Mexico; joel.mosqueda@uaq.mx

**Keywords:** rabbit, enteropathy, epizootic, fiber, fermentation

## Abstract

This study aimed to investigate the effect of feeding insoluble fiber on the microbiota and metabolites of the caecum and feces of rabbits recovering from epizootic rabbit enteropathy relative to non-infected rabbits. Rabbits that had either recovered from epizootic rabbit enteropathy or ones that had never had epizootic rabbit enteropathy were fed on a diet of 32% or 36% neutral detergent fiber until they were 70 days of age. At this point, the short-chain fatty acid and ammonia levels were measured in caecotroph and fecal samples and compared using 2 × 2 ANOVA. The microbial composition of the samples was also analyzed using next-generation sequencing and compared by PERMANOVA. Caecotrophic samples from previously affected rabbits on lower fiber diets had higher short-chain fatty acid contents and higher species diversity index values for some indices (*p* < 0.05), although the fecal samples showed lower species diversity levels (*p* < 0.05). In addition, the PERMANOVA analyses demonstrated that differences were detected in the microbial composition of both fecal and caecotrophic samples, depending on the disease status at the outset of the experiment (*p* < 0.05). The results of this work show that, although there is some potential in the use of high-fiber diets for the treatment of rabbits that have had epizootic rabbit enteropathy, they are not able to produce the same digestive tract properties as those seen in rabbits that have never had the condition. This is true even after the rabbits have recovered from epizootic rabbit enteropathy.

## 1. Introduction

Epizootic rabbit enteropathy (ERE) is a digestive disease that has negatively affected international rabbit production since the 1990s [1], with morbidities of up to 90% and mortalities of 80% [2]. ERE generally affects rabbits aged 3–7 weeks, reducing their daily food intake [3]. The earliest reports of ERE in Mexico were from late 2001 and early 2002 [4], and currently, ERE represents 32% of the digestive disease cases involved in rabbit production in Mexico, with the loss numbers of weaned rabbits being around 12–20% [5]. Although the etiology of ERE has not been fully described and identified so far, the most common treatment of ERE involves using oral antibiotics [6], indicating a probable bacterial origin. In keeping with the assumption that ERE has a bacterial basis, we have shown that several bacterial phyla are associated with this disease [7], although no single organism was identified as being the cause. However, since there is a trend towards avoiding antibiotics in rabbits’ diets, because there is a fear that overuse could help to promote resistant bacterial strains [8], this means that, in large outbreaks, the extensive use of the treatment could be problematic in the longer term. In addition, the inclusion of antibiotics in the food can lead to changes in the diversity of the normal gastrointestinal microbiota, causing a microbial imbalance, which can lead to dysbiosis [9]. The effects of antibiotics have already been investigated in rabbits, with different levels of effects being reported for different antibiotics [10,11], with some showing a clear impact on the microbial population. ERE is also characterized by a marked imbalance in the microbial community of the caecum [2,7], which, in turn, changes the profile of the metabolites produced and absorbed in the caecum [12]. Combining these pieces of information led to the hypothesis that an increase in the neutral detergent fiber (NDF) within the diet modulates the microbial communities and the metabolites they produce. This then has the potential to reduce the effects of ERE in rabbits during the fattening stage of their production for meat. In most nutritional studies with herbivores, the change in the dietary fiber content is assumed to impact mainly on the gut microbial community [13], although this, in turn, can influence the metabolites available for use by the host rabbit and can have additional benefits for the host, such as influencing the host’s anti-inflammatory signaling pathways [14].

While this increase in dietary fiber shows some promise as a means of impacting on the ERE levels, the underpinning microbiology behind it is less clearly understood. Therefore, the objective of this study was to compare the effect of low- (32% NDF) and high (36% NDF)-fiber diets, in conjunction with rabbits either having previously shown, but having now recovered from, ERE versus those that have never shown any clinical signs of ERE, to determine the impact on the microbial community and the metabolites they produce. This was performed by a combination of assessing short-chain fatty acid (SCFA) concentrations produced as a means of examining the metabolites arising from the digestion of food and the microbiota to evaluate the organisms involved in performing these roles.

## 2. Results

### 2.1. Body Weight Values

Statistical differences were observed in body weight by day (*p* < 0.001), disease (*p* < 0.001), day*disease (*p* < 0.05), and diet (*p* < 0.01); no effect was observed by diet*disease (*p* > 0.05) and day*disease*diet (*p* > 0.05). The initial weight of the rabbits that had had ERE was significantly lower (*p* < 0.001) than the weight of those that were healthy (Table 1). This observation continued throughout the experiment, even though the rabbits had recovered, with rabbits that had originally had ERE being significantly lighter than those that were healthy (*p* < 0.001). However, all the rabbits were gaining weight, and the daily weight gain was significantly higher in healthy rabbits (*p* < 0.01), principally with rabbits feeding being 36% fiber (*p* < 0.01).

Rabbits in groups T3 and T4 (the two groups containing animals that had ERE) presented with light (severity = 1) or moderate (severity = 2) diarrhea on the first day of the trial and during the subsequent 3 days but showed a reduction in symptoms over the next few days. By day 9, none of the rabbits had diarrhea or stool mucus, and no further symptoms were detected for the rest of the experiment. Groups T1 and T2 did not present diarrhea or signs associated with ERE during the duration of the experiment. The live weight values increased significantly in all groups from the start to the end of experimental period (*p* < 0.001).

### 2.2. Cecotrophic Fermentation

The results of the fermentation parameters are shown in Figure 1 and Table 2. No statistically significant differences were observed between the groups for the caecotroph pH values. In the case of the percentage of ammonia present, the values were not significantly different. However, there was a significant difference (*p* < 0.001) in the total production of SCFAs, with groups T2 and T3 having higher concentrations. Moreover, there were disease-dependent differences (*p* < 0.001), as well as differences in the interactions of the variables (*p* < 0.001).

### 2.3. Cecal Samples

A total of 846,594 DNA sequences were obtained, comprising 80 bacterial genera. This information is summarized in Table 3, with more detailed information contained in Appendix A. Following a rarefaction curve analysis, an almost horizontal asymptotic curve was obtained for almost all individuals (Figure 2), thereby indicating that operational taxonomic unit (OTU) detection had reached its peak.

Depending on the index studied, significantly higher levels of diversity were seen in the rabbits originally with ERE on the low-fiber diet relative to the caecotroph samples from the other groups. Both the Shannon and Chao1 indices showed significantly higher values (*p* = 0.022 and *p* = 0.020, respectively), but no such difference was seen for the ACE and Simpson values. This was true for the differences between diets (low- versus high-fiber), health status (healthy versus initially ERE-infected), and for the diet*health interactions in all cases. All indices for the caecotroph samples were highest in those collected from rabbits that initially had ERE while on low-fiber diets, although, in the case of the Simpson and ACE indices, these were not significantly higher.

### 2.4. Fecal Samples

A total of 1,152,002 DNA sequences were obtained, comprising 90 bacterial genera. This information is summarized in Table 4, with more detailed information contained in Appendix A. Following a rarefaction curve analysis, an almost horizontal asymptotic curve was obtained for almost all individuals (Figure 3), thereby indicating that OTU detection was reaching its peak.

Depending on the index studied, significantly different diversities were observed in the fecal samples. The Shannon indices showed no significant differences for comparisons between diets, the health status, or diet*health interactions. The Simpson diversity values were higher in healthy rabbits relative to the initially ERE-infected rabbits, as well as rabbits on higher fiber diets than on low-fiber diets, but no significant differences in the diet*health interactions were observed. No significant differences were detected for the ACE values for dietary differences or health status differences, although differences were detected for diet*health interactions, with healthy rabbits on a high-fiber diet showing a significantly higher index value than the other three groups. Finally, the Chao1 index was significantly higher for rabbits on high-fiber diets, but there was no significant difference based on the health status or diet*health interactions. In the case of the fecal samples, all indices were the highest for those from the healthy rabbits on high-fiber diets, although this was only significantly so for the ACE values.

### 2.5. Sequence Comparisons in Cecotroph and Fecal Samples

The most abundant phyla in all caecotroph groups were Firmicutes and Bacteroidetes (Table 5 and Figure 4). Typically, the most abundant families were *Ruminococcaceae*, *Bacteroidaceae*, the *Clostridiales_vadinBB60* group, and *Barnesiellaceae* (Appendix A).

However, there was a variation within the groups regarding the abundance of OTUs, and following PERMANOVA testing, there was a difference when comparing the healthy with the originally ERE-affected (*p* < 0.05). The individual dietary comparisons (*p* = 0.080) and health*diet interactions (*p* = 0.600) did not present significant differences.

The most abundant phylum in all fecal groups was Firmicutes (Table 5 and Figure 5). Typically, the most abundant families were *Ruminococcaceae*, *Verrucomicrobiaceae*, *Clostridiaceae,* and *Lachnospiraceae* (Appendix A). However, there was a variation within the groups regarding the abundance of OTUs, and following PERMANOVA testing, there were differences when comparing the healthy with the originally ERE-affected (*p* = 0.03) by diet (*p* = 0.01) and healthy versus ERE-infected (*p* = 0.03).

## 3. Discussion

A recent review of the factors associated with ERE and how to treat it [16] included the use of elevated fiber levels as one mechanism of treatment. This approach constituted the underpinning experimental design of this work. In addition to examining the fiber levels, the presence or absence of ERE at the outset of the experiment impacted on the productive parameters evaluated in this experiment. This effect of ERE is in agreement with a previous study [17], where it was reported that rabbits that survive ERE show lower weights compared to healthy rabbits in the same system. In the case of food consumption, groups with initial signs of ERE had a lower intake than healthy rabbits (data not shown). Those groups that were healthy at the outset, T1 and T2, had values in keeping with those reported previously for rabbits on diets with lower (≤30%) NDF [18], which are associated with a shorter retention time in the cecum and, in turn, an increase in food intake [19]. By way of contrast, the current work looks at the differences in the gut microbiota and associated metabolome in rabbits that have recovered from ERE versus those that have never suffered from it. As such, it is not an investigation of the causal agents of ERE but, rather, how closely rabbits respond and return to normality after an infection, together with the roles that different dietary fiber levels can promote. For this reason, we had an interest in examining both hard feces that are defecated and lost and also caecotrophs that are re-ingested, as the post-ERE recovery is poorly understood in both.

Rabbits that had ERE and were used during this trial had no more than level 2 symptoms. The reason for this was to maximize the likelihood that rabbits would recover from ERE and then survive to the end point of the experiment (70 day old), as those showing severe symptoms (level 3) are much less likely to survive, and our interests lay in examining the similarities between recovered rabbits and those that had never had symptoms. The effect that the diets had on diarrhea and mucus in feces is consistent with that reported previously [20], although the previous work involved a reduction of NDF from 30% to 25%. This produced an increase in the biodiversity of the microbiota in the ileum but reduced biodiversity in the caecum. In addition to this, a decrease in the presence of diarrhea (from 31% to 0.8%) was reported at day 20 in rabbits with ERE supplemented with 33.6% of NDF [21]. This was associated with a change in the caecotrophic ecosystem, leading to a higher production of volatile fatty acids [21] because of a change in the microbial profile of the caecum.

The concentrations of ammonia observed in groups T1 and T2 are similar to those obtained previously [22] with clinically healthy rabbits. These values are associated with a high absorption of ammonia from the intestinal mucosa of the rabbit [12]. Here, the ERE groups (T3 and T4) showed higher values than those reported in clinically healthy rabbits, although these differences were not significantly different. Differences might have been expected under these circumstances, as this could be associated with two different factors triggered by the ERE. First, there could have been damage to the intestinal mucosa, where the apoptosis of epithelial cells has been observed, together with atrophy, fusion, and destruction of the intestinal villi, and a reduced infiltration of inflammatory cells [23]. This resulted in the poorer absorption of digestive products into the bloodstream. Secondly, changes in the intestinal microbiota can also have an impact [2,24], with an increase in the abundance of bacteria of the genus *Clostridium* causing lesions in the intestinal mucosa due to the toxins generated [25], as well as an increase in ureolytic activity [21,25].

The indices for the diversity values observed are in keeping with a previous experimental work, where changes in the concentration of NDF led to changes in clinically healthy rabbits [4]. Regarding the changes in the richness of the microbiota, these agree with that reported in previous studies [2,24,26] for the ERE group that received the diet with 36% NDF. However, the genera affected in our study differ from those reported in previous studies [2,24]. The use of 36% NDF makes it is possible to manipulate the caecotrophic microbiota of rabbits with clinical ERE. This, in turn, may reflect a decrease in proteolytic activity and an increase in the hemicellulolytic communities of rabbits that are positive for ERE.

It has already been documented that the rabbit gut microbial community is different from that of other domesticated herbivores such as ruminants or horses [27,28], in addition to differences seen between different areas of the digestive tract of the rabbit [29]. In addition, previous next-generation sequencing approaches have shown differences between caecotrophic samples from four virulent (i.e., affected by ERE) versus two non-virulent rabbits [30]. In the current work, in terms of the population diversity of caecotrophic samples, the highest value of the Shannon Index (Table 3) was seen in the T3 group (rabbits that had originally had ERE and were fed on the low-fiber diet). This is particularly interesting, as the Shannon index places a high emphasis on species richness [27] but without the same emphasis on evenness that is seen in the Simpson index. This would suggest that there is still a very diverse population present but that the spread of the species has been affected. In keeping with this hypothesis, it would be anticipated that T3 would probably also have a higher Chao1 value, which measures numbers in the population. This is indeed the case (Table 3). These differences manifested at the level of a dietary consideration (*p* = 0.022 and *p* = 0.020 for Shannon and Chao1, respectively), ERE status (*p* = 0.004 and *p* = 0.013 for Shannon and Chao1, respectively), and the dietary*disease interaction (*p* = 0.013 and *p* = 0.024 for Shannon and Chao1, respectively).

It is also interesting to note that, when the Abundance-based Coverage Estimators (ACE) index is used, there is a difference between the samples from the rabbits that previously had ERE relative to those that had never had it. The ACE index is used to measure the number of rare OTUs detected [31], and the data in Table 3 suggest that there are differences in rare OTUs on the basis of whether a rabbit has had ERE or not. Moreover, there were no differences detected with the Simpson index, which looks at the abundance of dominant species [32], therefore suggesting that there has been no significant change in the number of abundant species. Thus, in some respects, the post-ERE microbial population is still different from that seen in rabbits that have never had the condition. Interestingly, the fecal samples did not show a similar pattern (Table 4), with higher Simpson Indices in the T1 and T2 samples (both groups have rabbits that have never suffered from ERE), indicating certain species are more dominant than others in this group relative to those that were seen in the rabbits that had originally suffered from ERE (*p* = 0.004). This pattern was also seen for rabbits with the higher fiber content in their diet (*p* = 0.002), with an increase in species dominance. However, this was not observed for a dietary*disease interaction (*p* = 0.279). Interestingly, the Chao1 value was also higher for the rabbits that never suffered from ERE (*p* = 0.033), and this is a measure of the number of species within the population [33]. This would suggest that, although there is a general level of dominance of the species, there are also a larger number of species present, possibly with some approaching trace levels. The largest Chao1 value is in T2 (rabbits that had never had ERE and were fed a 36% NDF diet), although this does not allow a significant difference to be seen in terms of dietary*disease interactions (*p* = 0.062). However, a significant difference in terms of this interaction was seen using the Abundance-based Coverage Estimator (*p* = 0.025), which is a measure of the detection of species above a certain minimum baseline [33]. This is despite not showing significant differences for either the diet (*p* = 0.092) or disease status (*p* = 0.082). However, no significant differences were observed for the Shannon Indices, which consider species richness [27].

Taken together, these results suggest differences between caecotrophic and fecal samples in terms of how they continue to be affected by the host rabbit having had ERE. In the caecotroph samples from rabbits that had previously had ERE and were fed on a lower fiber diet, increased species richness (Shannon Index) and diversity (Chao1) values without an increase in the dominance of certain species was shown. Conversely, the fecal samples showed an ERE effect when using the Simpson Index and a similar pattern for the Chao1 values. In addition, the higher fiber diet from rabbits that had never had ERE showed a higher level of low-abundance OTUs (Abundance-based Coverage Estimator) in fecal samples. In terms of identifying an individual species associated with the condition, as with other works, instead, we identified more than one organism that has been affected significantly in terms of its abundance even after the rabbit has recovered from ERE symptoms. What is clear though is that the differences detected in the sequence data from the caecotrophic samples do not translate to the equivalent information in fecal samples (Table 5). In this context, it is evident that there was no large-scale change to the organisms involved in causing ERE to persist at the level of the phylum. Clearly, any differences must be at a lower taxonomic level. It remains impossible to pinpoint a single organism as the causal agent, although one of the most interesting is probably *Akkermansia muciniphila,* which has a high abundance in the ERE positive groups in previous works. This is a microorganism that has been also associated with rabbits fed on lower fiber and that developed ERE [34], and we have previously reported this as an important microorganism related with ERE in Mexico [7]. It is a member of the phylum Verrucomicrobia, a phylum that showed no obvious patterns in terms of the abundance between the various samples collected in terms of caecotroph versus fecal, post-recovery from ERE versus never having been infected, and the two levels of fiber in the diet (Table 5). Moreover, *Akkermansia* was the only genus observed within the sequences identified as belonging to the phylum Verrucomicrobia (Appendix A).

Another organism of interest from our previous work [7], *Cloacibacillus porcorum* (which is a member of the phylum Synergestes) was not detected in the current work as the only genus detected here from the phylum Synergestes was Synergestes (Appendix A). Members of this phylum were present at very low levels in all fecal samples (irrespective of their ERE status) and caecotroph samples that recovered from ERE but not in those that never had ERE. However, specifically for *Cloacibacillus porcorum*, its absence in the current work may be an indication that it has been lost following recovery from ERE in those rabbits that previously had it.

The final group of organisms that we highlighted as a potential causal agent for ERE were members of the genus Clostridium. Although the Firmicutes were present as one of the most abundant phyla (in some cases, the most abundant phylum) in all the samples, this was due primarily to the high abundance of members of the Class *Ruminococcaceae*. In terms of members of the genus Clostridium, these were only present at very low levels in the fecal samples and in the caecotrophs collected from previously ERE rabbits on the low-fiber diet. Furthermore, due to the wide range of species of bacteria within the genus Clostridium, these cannot be assumed to be the same species as were identified in the previous work.

Therefore, none of the organisms we highlighted in our previous work [7] as being potential causal agents for ERE could be detected in the samples here, even for rabbits that had recovered from ERE. This means that, within the limits of detection here, those organisms were not present in rabbits that had recovered from ERE.

In the wider context, our data identify similar levels of Bacteroides but lower levels of Firmicutes than were seen in the previous work for both fecal and cecal samples, e.g., [35], although these authors used a much lower fiber content in the diet (18.7%) than either of our sets of conditions. However, it is worth noting that, in our previous work [7], we observed a lower abundance of Firmicutes than has been published previously, which we speculated may be, at least in part, be due to the removal of a PCR step prior to sequencing, as the PCR stages can introduce a bias in terms of amplification. The values we see here for Firmicutes in the fecal samples are in keeping with those we reported previously. However, the caecotroph values are lower than that, although similar to those seen in wild rabbits [29]. Due to the differences detected in terms of Firmicutes relative to the other work, this makes any inferences to the Firmicutes:Bacteroides ratio or the impact SCFA production relative to the Firmicute numbers difficult to achieve.

Interestingly, members of the phylum Tenericutes were only observed in the cecum of rabbits that had never suffered from ERE and were absent from all fecal samples. The detection of Tenericutes in cecal samples of rabbits, with an absence in fecal samples, has been reported previously, although Tenericutes were found in fecal samples from guinea pigs in the same work [29]. This differs from the observations in another work [35], which detected Tenericutes in both fecal and cecal samples from rabbits at 8.17% and 7.48%, respectively.

To date, the roles of many of the bacteria identified in the digestive tract of rabbits remain unclear. This is true both in terms of their function in healthy rabbits and also in those that are either currently suffering from ERE or have previously suffered from ERE, as well as in response to different levels of dietary fiber. For example, members of the Class Synergistia (specifically the genus *Synergistes*) were detected at low levels in fecal samples in groups of rabbits but only in cecal samples from rabbits that had previously suffered from ERE, irrespective of their dietary fiber content (Appendix A). This is interesting in the context of a previous work that reported that this group of organisms increased in abundance in the cecum of rabbits that were exposed to zearalenone, a mycotoxin that is known to affect the structure of the cecal microbiota [36]. Thus, this could be speculated to be an example of organisms that may act as stress indicators, although this needs further investigation.

## 4. Materials and Methods

### 4.1. Experimental Design

The experimental work was carried out at the Amazcala Campus of the College of Natural Sciences, Autonomous University of Queretaro. Ethical conditions were approved using criteria established by the Ethics Advisory Committee for Animal Experimentation, University of Zaragoza [37]. The experimental protocol was approved by the Bioethics Committee of the College of Natural Sciences, Autonomous University of Queretaro (number 93FCN2016). Sixteen female New Zealand rabbits aged 32 days old were housed as groups of 4 rabbits in American-type cages (90 cm × 60 cm × 40 cm) equipped with feeders and drinking fountains. Rabbits were allocated in a 2 × 2 design experiment, with 4 rabbits per group. All rabbits that had ERE at the outset acquired it spontaneously (i.e., it was not induced) and had not been given any antibiotic treatment for the condition. An ERE diagnosis was confirmed by the first author of this work, who is a qualified veterinary surgeon. At the outset of the experimental work (i.e., when the rabbits were 32 days old), the rabbits were allocated to a group based on either having ERE or being clear of it (deemed healthy) and on a diet of 32% NDF, which is recommended for the intestinal health of the rabbit [38], or 36% NDF, which has been suggested for the treatment of ERE [26]. Although it was not possible to balance rabbits for weight between the ERE and healthy groups, due to the ERE rabbits being significantly lighter at the outset, the rabbits were balanced as closely as possible for the dietary regimes. The diets were formulated with the same ingredients: alfalfa meal, canola meal, wheat bran, sunflower meal, sodium chloride, calcium carbonate, vitamins, and minerals (Table 6). No antibiotics or anticoccidial drugs were included in the diet. All rabbits were given ad libitum access to food and water.

The experimental groups were designated as follows: T1 = healthy with 32% NDF, T2 = healthy with 36% NDF, T3 = ERE clinical signs with 32% NDF, and T4 = ERE clinical signs with 36% NDF. The presence of diarrhea (an ERE clinical sign) was checked by measuring the severity (0 = no diarrhea, 1 = slight, 2 = moderate, and 3 = severe). At the start of the experimental work, all rabbits categorized as having ERE were around the 1 to 2 boundaries of severity. Information regarding the productivity values that were recorded or measured were: initial weight (IW), weekly weight gain, final weight (FW), food consumption (FC), and daily weight gain (DWG).

### 4.2. Sample Collection

At 70 days of age, samples of caecotrophs were collected from all rabbits. Collections were achieved by temporarily (07:00 a.m. to 11:00 a.m.) moving rabbits to individual cages and placing polyurethane conical collars (6-cm narrow diameter and 27-cm wide diameter) on the necks of the rabbits to prevent them ingesting their caecotrophs. The caecotrophs were identified on the basis of their consistency and appearance; they were expelled in the form of clusters with a shiny appearance. During this collection period, a sample of fecal waste was also collected, with these being identified as dry and rough, and expelled as individual pellets. The pH of the caecotrophs was measured following a collection of 2 g samples and mixing them with 5 mL of distilled water.

### 4.3. Short-Chain Fatty Acid Analysis

For the determination of the SCFA content, 2 g of the sample were collected in a 15 mL conical centrifuge tube with 5 mL of HPLC-grade acetone and 1 mL of HPLC-grade water. This was mixed by vortexing for 15 s, followed by the addition of 100 μL of 85% (*w*/*v*) phosphoric acid, and the solution was mixed again by vortexing for 10 s and centrifuged for 23 min at 21,000× *g* at 4 °C; the supernatant was separated by passing through a 0.20 μm filter. The filtrate was subsequently placed in amber vials with a lid to carry out the liquid chromatography analysis and stored at −80 °C until used. Liquid chromatography was performed using an Agilent 6890 gas chromatograph with a flame ionization detector and a DB-FFAP 30 m × 0.25 mm × 0.25 μm column (Agilent Technologies, Wilmington, NC, USA) for the analysis of the filtered supernatant with a 0.2 μm syringe filter [40]. The conditions followed a split mode (20:1) with an inlet temperature of 220 °C and pressure of 168 kPa. A volume of 1 μL was injected, with a constant flow of 1.4 mL/min, using helium as a carrier. The detector temperature was 250 °C. The column initially operated at 35 °C, with a hold period of 30 s, before increasing at 10 °C/min until 90 °C was reached. This temperature was held for 2 min and was followed by an increase of 12 °C/min until 230 °C was reached, and this was held for 6 min.

### 4.4. Ammonia Analysis

For the analysis of the ammonium content, a Kjeldahl approach was used [39,41]. Initially 2 g of caecotrophs were placed in a Kjeldahl flask to which 200 mL of distilled water at 20 °C was added, followed by the addition of 4 glass boiling beads of 3 mm diameters (PROLAB, Tlajomulco de Zuñiga, JAL, MEX) and 2 g of MgO. This was then distilled using a recovery rate of 99.5%. The solution was neutralized by the addition of 20 mL 0.05 M sulfuric acid for 4 min. The products of the distillation process were collected in 500 mL Erlenmeyer flasks. To each flask, 100 mL of 2% (*w*/*v*) H_3_BO_3_ and 3 drops of indicator solution (composed of 20 mL of 0.05% (*w*/*v*) methyl red and 4 mL of 0.2% (*w*/*v*) methylene blue) were added to ensure the solution was neutral. Titrations were performed using 0.1 M H_2_SO_4_, which was dripped with a graduated pipette until the color of the indicator solution changed. These values were compared against a blank (i.e., without caecotrophs), and the N% of the ammonia value was calculated using the equation below:N% = [(A–B) * 0.014 * N * 100]/M(1)
where: A and B are the volumes of 0.1 M H_2_SO_4_ (mL) used in the titration of the caecotroph and control samples, respectively, N is the normality of H_2_SO_4_, and M is the weight of the caecotroph sample measured in grams.

### 4.5. DNA Extraction and Next-Generation Sequencing

The rest of the caecotrophs were placed in 1.5 mL tubes that were free of both RNases and DNases and immediately frozen in liquid nitrogen. They were then transported to the Veterinary Microbiology Laboratory of the College of Natural Sciences, Autonomous University of Queretaro, where they were stored at −80 °C. The DNA was extracted from caecotrophs using Qiagen tool kits with silica and micro pearl columns, according to the manufacturers’ protocols. All caecotrophic and fecal samples (i.e., 16 caecotrophic and 16 fecal samples) were used for the sequence analysis.

Next-generation DNA sequencing of the V3–V4 region of the 16S rRNA gene was performed on an Illumina MiSeq^®^ System by the Sequencing and Identification Unit, National Institute of Genomic Medicine, Mexico City, Mexico. The sequencing process consisted of amplicon processing, Ampure beads purification, nucleic acid quantification by fluorescence, automated chip electrophoresis, and loading onto the Illumina MiSeq system. The library preparation was performed by the bridge method with the Nextera™ Flexible DNA Library Prep Kit (Illumina Inc. USA, San Diego, CA, USA). Sequences were filtered with Trimmomatic [42] to remove low-quality reads, followed by removal of the sequencing adapters, leaving reads of ~480 bp. A further quality inspection was performed with FastQC [43]: involving trimming, filtering, identifying unique sequences, constructing tables of operational taxonomic units, removing chimeric sequences, assigning taxonomic identification, determining the abundance and diversity using the R software Project, and the specialized library DADA2 [44].

### 4.6. Statistical Analysis

The data for short-chain fatty acids and ammonia were analyzed using a 2 × 2 ANOVA factorial design, using IBM SPSS Statistics for Windows (Version 19.0. Armonk, NY: IBM Corp, Armonk, NY, USA). In the case of values that showed significant differences, Tukey’s post hoc analyses were performed. Data for live weight relative to disease status and the fiber content were compared by repeated measurements two-way ANOVA with replicates (*n* = 4). The use of 4 replicates per treatment is more than many previous publications have used for similar experiments across a range of different works with a range of animal species (e.g., *n* = 3 being used regularly), although not as large as some works that use larger numbers [35,45].

In addition, PERMANOVA was carried out following the methodology described in the manual of the Vegan package [45]. In the case of the analysis of the bacterial diversity, this was assessed by calculating the Shannon Index, the Simpson Index, the Chao1 Index, and the Abundance-based Coverage Estimators (ACE), following the methodology of Oksanen et al. [46]. The data obtained were compared using a 2 × 2 factorial design ANOVA using IBM SPSS Statistics for Windows (Version 19.0, Armonk, NY, USA: IBM Corp). A rarefaction curve was produced with the rarefy script: Rarefaction Species Richness in Vegan: Community Ecology Package [46].

## 5. Conclusions

In conclusion, even though the rabbits that had previously suffered from ERE made a full recovery in terms of no longer having diarrhea, they were still smaller than animals that never had it and have a different gut microbiome and metabolome produced by these organisms.

## Figures and Tables

**Figure 1 pathogens-11-00571-f001:**
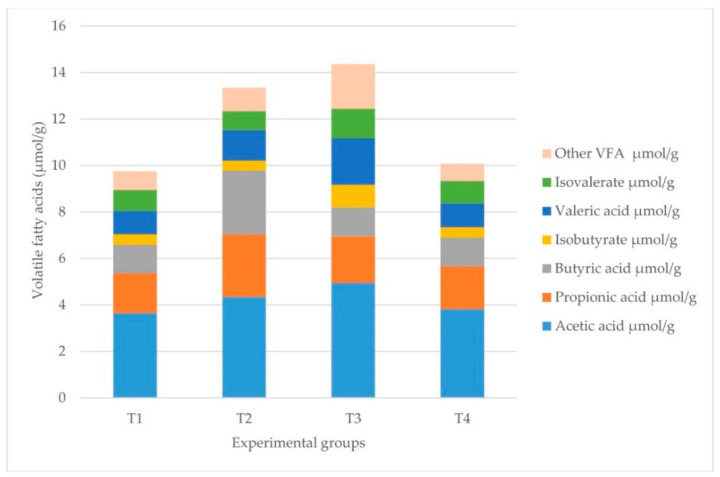
Volatile fatty acid (VFA) profile per group in µmol/g. T1 = healthy rabbits with a 32% NDF diet; T2 = healthy rabbits with a 36% NDF diet; T3 = rabbits which had ERE at the outset on a diet with 32% NDF; and T4 = rabbits which had ERE at the outset with a 36% NDF diet.

**Figure 2 pathogens-11-00571-f002:**
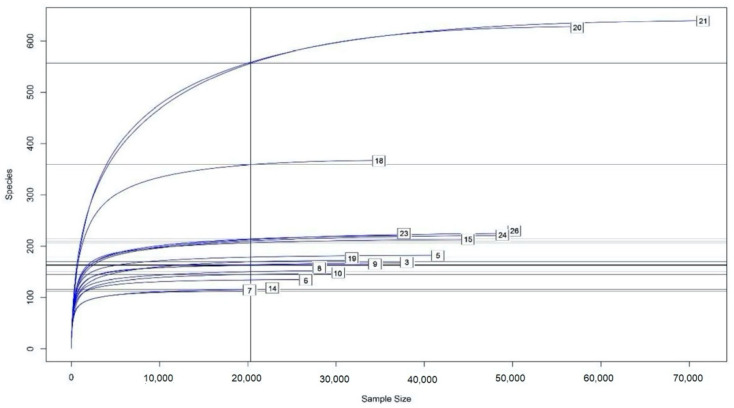
Rarefaction curve for sequence analysis of caecotrophs. The vertical axis shows the number of different species (operational taxonomic units, OTUs) that would be expected to be found after sampling the number of sequences shown on the horizontal axis. The curvature towards the horizontal indicates the greater sequencing effort required to observe new species. The curve was obtained with the Rarefaction Species Richness script from the Vegan package on the RStudio platform [15]. Group identification: T1 (healthy rabbits on a diet with 32% NDF): 3, 5, 6, and 7. T2 (healthy rabbits on a diet with 32% NDF): 8, 9, 10, and 14. T3 (rabbits that had ERE at the outset on a diet with 32% NDF): 15, 18, 19, and 20. T4 (rabbits that had ERE at the outset on a diet with 36% NDF): 21, 23, 24, and 26.

**Figure 3 pathogens-11-00571-f003:**
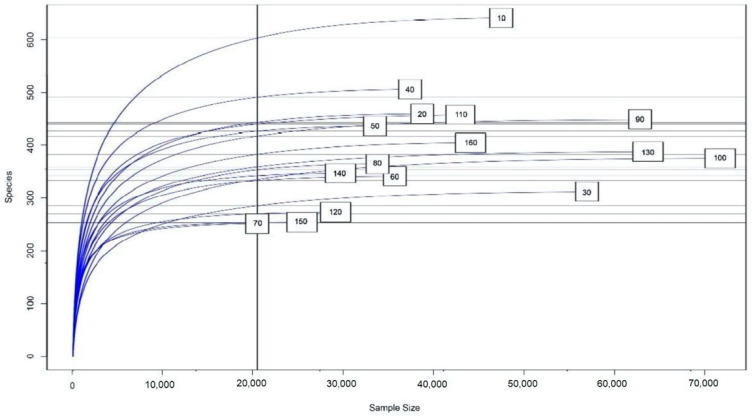
Rarefaction curve for the sequence analysis of feces. The vertical axis shows the number of different species (operational taxonomic units, OTUs) that would be expected to be found after sampling the number of sequences shown on the horizontal axis. The curvature towards the horizontal indicates the greater sequencing effort required to observe new species. The curve was obtained with the Rarefaction Species Richness script from the Vegan package on the RStudio platform [15]. Group identification: T1 (healthy rabbits on a diet with 32% NDF): 10, 20, 30, and 40. T2 (healthy rabbits on a diet with 32% NDF): 50, 60, 70, and 80. T3 (rabbits that had ERE at the outset on a diet with 32% NDF): 90, 100, 110, and 120. T4 (rabbits that had ERE at the outset on a diet with 36% NDF): 130, 140, 150, and 160.

**Figure 4 pathogens-11-00571-f004:**
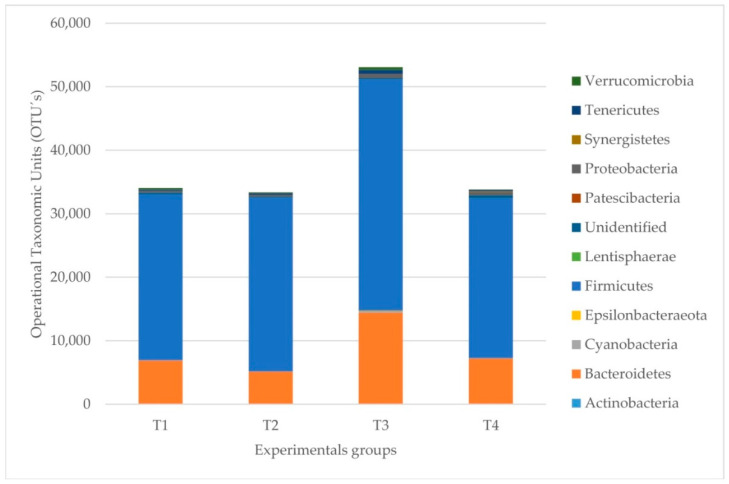
Microbial profile of the abundances of the cecal samples by groups. The most abundant phyla are shown in each experimental group. T1 = healthy rabbits with a 32% NDF diet. T2 = healthy rabbits with a 36% NDF diet, T3 = rabbits that had ERE at the outset on a diet with 32% NDF, and T4 = rabbits that had ERE at the outset on a diet with 36% NDF.

**Figure 5 pathogens-11-00571-f005:**
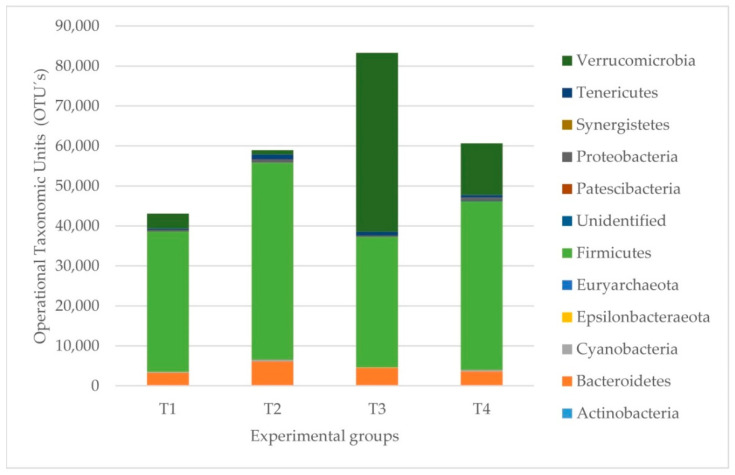
Microbial profiles of the abundances of the fecal samples by groups. The most abundant phyla are shown in each experimental group. T1 = healthy rabbits with a 32% NDF diet, T2 = healthy rabbits with a 36% NDF diet, T3 = rabbits that had ERE at the outset on a diet with 32% NDF, and T4 = rabbits that had ERE at the outset on a diet with 36% NDF.

**Table 1 pathogens-11-00571-t001:** Mean body weights (grams) for each group of rabbits at different time points. T1 = healthy rabbits on a diet with 32% NDF, T2 = healthy rabbits on a diet with 36% NDF, T3 = rabbits which had ERE at the outset on a diet with 32% NDF, and T4 = rabbits that had ERE at the outset on a diet with 36% NDF. SEM values are shown in parentheses.

	Day										
Treatment	0	3	6	9	12	15	18	21	25	30	35
**T1** (*n* = 4)	636	733	834	938	1032	1122	1247	1366	1556	1819	2095
	(49.6)	(43.1)	(44.7)	(44.7)	(35.8)	(20.7)	(34.8)	(41.8)	(46.1)	(41.9)	(33.6)
**T2** (*n* = 4)	621	730	843	959	1040	1143	1265	1389	1580	1865	2193
	(52.9)	(60.6)	(29.0)	(31.9)	(25.5)	(37.7)	(34.1)	(17.5)	(23.3)	(33.2)	(35.7)
**T3** (*n* = 4)	427	507	585	680	771	859	954	1089	1320	1532	1843
	(54.0)	(70.1)	(48.9)	(40.2)	(38.5)	(40.8)	(42.8)	(57.2)	(54.7)	(28.0)	(50.3)
**T4** (*n* = 4)	474	565	645	733	825	916	1015	1209	1348	1582	1927
	(44.1)	(43.0)	(34.8)	(37.9)	(30.0)	(24.5)	(13.0)	(24.3)	(49.7)	(35.0)	(39.0)
*n* = 16											

**Table 2 pathogens-11-00571-t002:** Volatile fatty acid, ammonia, and pH values for the caecotroph samples collected. Mean and standard deviation values are shown. T1 = healthy rabbits on a diet with 32% NDF. T2 = healthy rabbits on a diet with 36% NDF. T3 = rabbits which had ERE at the outset on a diet with 32% NDF. and T4 = rabbits that had ERE at the outset on a diet with 36% NDF. NS denotes values that were not significantly different.

Metabolite/Parameter	T1	T2	T3	T4	*p*-Value—Diet	*p*-Value—Disease	*p*-Value—Interaction
Total VFA μmol/g	9.75± 0.14	13.35 ± 0.16	14.36 ± 0.19	10.08 ± 0.16	0.034	0.001	<0.001
Acetic acid μmol/g	3.65 ± 0.10	4.35 ± 0.01	4.95 ± 0.09	3.82 ± 0.10	0.028	0.001	<0.001
Propionic acid μmol/g	1.71 ± 0.09	2.68 ± 0.03	2.00 ± 0.10	1.85 ± 0.08	0.007	<0.001	<0.001
Butyric acid μmol/g	1.24 ± 0.03	2.76 ± 0.02	1.25 ± 0.02	1.24 ± 0.02	<0.001	<0.001	<0.001
Isobutyrate μmol/g	0.43 ± 0.02	0.41 ± 0.01	0.97 ± 0.02	0.43 ± 0.02	<0.001	<0.001	<0.001
Valeric acid μmol/g	1.02 ± 0.01	1.32 ± 0.01	2.00 ± 0.04	1.03 ± 0.02	<0.001	<0.001	<0.001
Isovalerate μmol/g	0.90 ± 0.01	0.82 ± 0.01	1.27 ± 0.02	0.97 ± 0.01	<0.001	<0.001	<0.001
Ammonia %	0.06 ± 0.01	0.04 ± 0.01	0.16 ± 0.07	0.07 ± 0.01	NS	NS	NS
pH	6.58 ± 0.17	6.46 ± 0.07	6.43 ± 0.06	6.43 ± 0.13	NS	NS	NS

**Table 3 pathogens-11-00571-t003:** Summary of the sequences identified and comparison of the microbial diversity indices in cecal samples. Mean values are shown, with standard deviations shown in parentheses. T1 = healthy rabbits on a diet with 32% NDF, T2 = healthy rabbits on a diet with 36% NDF, T3 = rabbits which had ERE at the outset on a diet with 32% NDF, and T4 = rabbits that had ERE at the outset on a diet with 36% NDF.

	T1	T2	T3	T4	*p*-Value—Diet	*p*-Value—Disease	*p*-Value—Interaction
Number of Sequences	164,382	259,136	272,098	150,978			
Number of Operational Taxonomic Units (OTU)	467	462	767	471			
Number of Identifiable genera	33	35	72	35			
Shannon Index	4.66 (0.53)	4.89 (0.54)	10.02 (3.05)	5.38 (1.14)	0.022	0.004	0.013
Simpson Index	4.01 (0.24)	4.02 (0.28)	4.33 (0.28)	4.07 (0.21)	0.349	0.165	0.312
Abundance-based Coverage Estimators (ACE)	0.40 (0.40)	0.56 (0.32)	1.42 (0.76)	1.23 (0.67)	0.963	0.011	0.544
Chao1 Index	165 (34.4)	159 (29.5)	464 (205.2)	178 (55.0)	0.020	0.013	0.024

**Table 4 pathogens-11-00571-t004:** Summary of the sequences identified and comparison of the microbial diversity indices in fecal samples. Mean values are shown, with standard deviations shown in parentheses. T1 (healthy rabbits on a diet with 32% NDF) = healthy rabbits on a diet with 32% NDF, T2 (healthy rabbits on a diet with 32% NDF) = healthy rabbits on a diet with 36% NDF, T3 (rabbits that had ERE at the outset on a diet with 32% NDF) = rabbits showing symptoms of ERE on a diet with 32% NDF, and T4 (rabbits that had ERE at the outset on a diet with 36% NDF) = ERE rabbits showing symptoms of ERE on a diet with 36% NDF.

	T1	T2	T3	T4	*p*-Value—Diet	*p*-Value—Disease	*p*-Value—Interaction
Number of Sequences	256,412	268,214	384,230	243,146			
Number of Operational Taxonomic Units (OTU)	818	1018	842	858			
Number of Identifiable genera	63	64	66	58			
Shannon Index	9.11 (1.53)	11.50 (1.21)	9.61 (0.38)	9.74 (1.28)	0.055	0.306	0.081
Simpson Index	4.02 (0.25)	4.58 (0.19)	2.82 (0.87)	3.92 (0.16)	0.004	0.002	0.279
Abundance-based Coverage Estimators (ACE)	0.66 (0.30)	1.87 (0.87)	0.84 (0.51)	0.64 (0.36)	0.092	0.082	0.025
Chao1 Index	375 (105.6)	575 (103.3)	399 (38.4)	415 (93.7)	0.033	0.155	0.062

**Table 5 pathogens-11-00571-t005:** Percentage abundance of the major phyla based on a next-generation sequence analysis; numbers in parentheses are SEM values. T1 = healthy rabbits on a diet with 32% NDF, T2 = healthy rabbits on a diet with 36% NDF, T3 = rabbits that had ERE at the outset on a diet with 32% NDF, and T4 = rabbits that had ERE at the outset on a diet with 36% NDF. Within any individual row, entries that share the same superscript are not significantly different from each other. Rows where no superscripts are present indicates that all values are significantly different from each other.

	Cecal Samples				Fecal Samples			
Phylum	T1	T2	T3	T4	T1	T2	T3	T4
Actinobacteria	-	-	0.007 (0.007) ^a^	-	0.04 (0.02) ^b^	0.04 (0.01) ^b^	0.08 (0.03) ^b^	0.04 (0.02) ^b^
Bacteroidetes	10.2 (0.22) ^ab^	8.8 (0.28) ^a^	12.0 (0.59) ^b^	10.7 (0.60) ^ab^	5.6 (0.59) ^c^	6.7 (0.16) ^d^	6.0 (1.28) ^cd^	5.2 (1.23) ^cd^
Cyanobacteria	8.0 (0.54) ^ab^	7.3 (0.25) ^a^	10.2 (0.93) ^b^	7.7 (1.08) ^ab^	5.3 (0.61) ^a^	6.3 (0.13) ^b^	5.8 (1.32) ^ab^	4.9 (1.09) ^ab^
Epsilonbacteraeota	-	-	0.004 (0.002)	-	0.007 (0.007)	0.013 (0.008)	-	0.008 (0.008)
Euryarchaeota	-	-	-	-	-	0.011 (0.008)	0.002 (0.002)	0.004 (0.002)
Firmicutes	49.2 (1.6) ^a^	55.9 (2.1) ^a^	43.3 (0.8) ^a^	47.0 (2.9) ^a^	67.6 (3.7) ^b^	61.0 (0.89) ^c^	64.8 (7.8) ^bc^	69.4 (7.0) ^bc^
Lentisphaerae	0.004 (0.004)	-	0.002 (0.002)	-	-	-	-	-
Not Identified	10.2 (0.2) ^ab^	8.7 (0.3) ^a^	12.0 (0.6) ^b^	10.7 (0.6) ^ab^	5.3 (0.6) ^c^	6.3 (0.1) ^d^	5.7 (1.3) ^cd^	4.9 (1.1) ^cd^
Patescibacteria	3.6 (1.3) ^a^	3.2 (1.0) ^a^	7.0 (1.7) ^a^	3.4 (1.9) ^a^	0.003 (0.003) ^b^	-	0.008 (0.008) ^b^	0.010 (0.010) ^b^
Proteobacteria	7.9 (0.9) ^ab^	6.4 (0.8) ^a^	7.7 (1.5) ^b^	8.8 (1.1) ^ab^	5.3 (0.6) ^c^	6.5 (0.2) ^d^	5.8 (1.3) ^cd^	5.1 (1.1) ^cd^
Synergistetes	-	-	0.013 (0.013) ^a^	0.199 (0.118) ^a^	0.003 (0.003) ^b^	-	0.011 (0.006) ^b^	0.005 (0.005) ^b^
Tenericutes	6.8 (1.2)	5.8 (1.0)	4.9 (1.4)	7.3 (1.5)	5.5 (0.6)	6.6 (0.2)	5.9 (1.3)	5.2 (1.2)
Verrucomicrobia	4.2 (0.9) ^abc^	4.0 (0.8) ^abc^	3.1 (0.8) ^abc^	4.2 (1.1) ^abd^	5.4 (0.61) ^de^	6.6 (0.15) ^abcd^	5.9 (1.29) ^abcd^	5.2 (1.20) ^acbcd^

**Table 6 pathogens-11-00571-t006:** Proximal chemical compositions of the diets. Data were obtained from the Animal Nutrition Laboratory, Autonomous University of Querétaro following the AOAC manual [39].

Proximal Chemical Analysis	Diet 1 (32%)	Diet 2 (36%)
Dry weight	91.2	91.3
Neutral Detergent Fibre (NDF)	31.7	35.8
Acid Detergent Fibre (ADF)	23.5	25.6
Hemi cellulose	8.2	10.2
Crude Protein	15.8	16.0
Ash	8.8	8.6
Lignin	7.8	7.8
Fat Content	1.7	1.8
Phosphorus	0.85	0.86
Calcium	1.42	1.43
Selenium	0.35	0.35
Vitamin A UI/kg	5167	5147
Vitamin D3 UI/kg	974	974
Vitamin E mg/kg	53.3	55.2
Vitamin B12 mg/kg	4.3	4.5

## Data Availability

Not applicable.

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
