# Peer review of "Effect of Feeding Insoluble Fiber on the Microbiota and Metabolites of the Caecum and Feces of Rabbits Recovering from Epizootic Rabbit Enteropathy Relative to Non-Infected Rabbits"

_pathogens, 2022, doi:10.3390/pathogens11050571_

Round 1
Reviewer 1 Report
The manuscript is well written and addresses a very interesting topic.
Small questions should be improved before being published:
1) Material and methods section: add more details on the statistical part abour data pre-treatment
2) Table 4: review the superscripts.
3) Discussion: perfomer a deeper discussion in relation to the bacteria found in the study and their relationship with the health/dysbiosia of the animal, not only on Firmicutes, Bacteroides and their ratio. For example, what role can Patescibacteria, Synergistetes and Tenericutes play on healthy or susceptibily to ERE ?
Reviewer 2 Report
In this article, the authors well described and discussed both the effect of the administration of different percentages of insoluble fiber on the microbiota and metabolites of the caecum and feces of rabbits recovering from epizootic rabbit enteropathy. The topic is important and has a novelty in the point of the continuous growing awareness of the efficacy of novel therapeutic candidates for the targeted treatment of such disease in the place of antibiotics. In fact, ERE is a severe gastrointestinal syndrome disease that can cause substantial economic losses beyond causing serious discomfort to the animals.
For this reason, I recommend the publication of the manuscript, provided that some minor revisions are carried out:
- authors should better discriminate between the use of “microbiome” or “microbiota”, as the first is supposed to be related to the genome part of the microbial community, which is called microbiota.
- please delete p>0.05 since it is useless
- bacterial family names should be in italic
- line 108: delete one parenthesis
- materials and methods section should be subdivided into more subsections to improve the readability
- were antibiotics administered before the investigation period to treat or prevent pathologies?
- ad libitum should be in italic
Reviewer 3 Report
Effect of feeding insoluble fiber on the microbiome and metabolites of the caecum and feces of rabbits recovering from epizootic rabbit enteropathy relative to non-infected animals
A brief summary
In this manuscript, a potential diet for rabbits recovering from ERE is described in detail. The data could be useful for further veterinary applications.
Broad comments
The main strength of the study is the detailed analysis of ERE. The main weakness is the statistical analysis and the low sample size.
Specific comments
Title and abstract:
Page 1, Line 25 and throughout the manuscript: authors should use “Rabbits”, instead of “Animals”.
Introduction:
Page 2, Line 50: …the etiology of ERE has not been fully described and identified so far, …
Results:
Page 2, Line 82:
The changes of body weights should be a subsection, 2.1. The effect of time has to be analyzed as well.
Page 8, Table 4:
The significances are not explained, a, b, c, ab, abcd, etc. It is not clear, what those superscripts mean.
Materials and methods:
Materials and methods should be divided to subsections, e.g., Animals, Experimental design, Statistical analysis, etc.
Page 13, line 175: The sex of the rabbits is missing.
Page 13, line 177: It is the major weakness of the manuscript. 4 rabbits/group is a low sample size for the either two-way or three-way ANOVA. The minimum sample size/group is 5.
Sixty rabbits were used in a study with similar diets and experimental design (reference [33] in this manuscript). Another study used 40 rabbits (10/diet) for faecal digestibility trial (Garcia et al., 2020, https://doi.org/10.3390/ani10081346 ).
Page 14, line 192: Were the rabbits of the T3 and T4 groups infected experimentally or spontaneously? Were they diagnosed with ERE by a local veterinarian?
Page 15, line 244: two-way ANOVA is an adequate statistical probe, if the effects of two factors and their interactions are analyzed. But in case of the body weights, the effect of another factor, time has to be also analyzed (Results, Figure 1.). In that case, it is a three-way repeated measures ANOVA, where the three factors are: disease, diet, time.
Page 15, line 246. The exact type of the post hoc test has to be added.
Round 2
Reviewer 3 Report
Page 15, line 244: There is repeated measures ANOVA (GLM -> Repeated measures in SPSS). In your experimental schedule (Table 1), you have 11 time points, and 4 groups. Without conducting repeated measures ANOVA, you cannot evaluate the efffect of time on body weigths.
Page 2-3, Table1. According to the revised version, 3 way ANOVA performed, but the effect of time, disease and diet on body weight and their interactions are still missing.
Author Response
Please see the attachment.

This manuscript is a resubmission of an earlier submission. The following is a list of the peer review reports and author responses from that submission.
Round 1
Reviewer 1 Report
Manuscript ID: animals-893384
Title: Effect of feeding insoluble fiber on the microbiome and metabolites of the cecum and feces of rabbits recovering from epizootic rabbit enteropathy relative 4 to non-infected animals
General comments to manuscript
The ERE is an important digestive disease in rabbits which causes a high mortality. This manuscript presents an interesting topic like study of increasing neutral detergent fiber in the dietary of rabbits in order to decrease the number of cases of ERE.
I consider that this manuscript falls within the scope of this journal and recommend its publication but with major review.
Abstract
Please, remember the Abstract is a single paragraph that includes the objectives of study and summarizes the results in an understandable form using statistical evidence (P-values). Abbreviations are defined at first use in the ABSTRACT and again in the body of the manuscript. Authors have to include a conclusion at the end of this section.
The authors make a very long introduction. Please, make a resume of lines 34 to 41.
Line 41: Please, insert “The objective of this study was to investigate the effect of different levels …………….
Line 43: Authors use inappropriately the term as “cecal samples”. Note that cecal samples are taken in cecum (after slaughter to animal), but this study analysed the cecotrophs (why do authors assume that cecotrophs are identical to cecal samples?). Please, authors have to correct this term a long all text or to explain why both terms are similar for authors.
Authors must indicate the age of animal.
The results are very generally “ Cecal samples from previously affected animals on lower fiber diets had higher volatile fatty acid content and higher species diversity index values, although fecal samples showed lower species diversity levels (Please, exactly in which VFAs and genera did they diverge?). In addition, PERMANOVA analyses demonstrated that differences were detected in the microbial composition of both fecal and cecal samples depending on the disease status at the outset of the experiment (Please, give values to remark the differences in composition between the different disease status).
Introduction
The introduction should end with a sentence that begin as "the objective of this study was ....
Materials and Methods
This section should be rewritten and provide more details on group classification, collection of samples, and the model used.
For example:
Pag Line
3 98-101 The groups were established in accordance with the presence of ERE symptoms in the animal, but when? at the beginning or at the end of the experiment, that is at 32 days or at 70 days?.
For analysing the data (VFAs, pH ammonia, diversity, and abundance), did the model include the classification of ERE at the beginning or end of the experiment?
Pag line
3 105-108 When were the hard feces collected? Were the animals fasting?
More details about the model used should be given, indicating carefully all the fixed effects, and specifying their levels.
Results
Pag Line
4 161-165 Authors could give more details about the differences for weight and daily weight between groups. Please, indicate the values.
4 171-174 Rabbits in groups T3 and T4 presented with light (severity = 1) or moderate (severity = 2) diarrhoea on the first day of the trial and during the subsequent 3 days but showed no further symptoms for the rest of the experiment. Diarrhoea and stool mucus in both treatments with ERE decreased from the fourth day of treatment and disappeared by the ninth day ". please, indicate in M&M how were diarrhoea and stool mucus symptoms check? And when …at 0d, 1d, 2d, 3d and 9d or more?
4 179-179 If there is no difference for ammonia (P > 0.05), authors have to delete “In the case of the percentage of ammonia the greatest value was seen in group T3”
4 181-182 Please, provide more information for each VFA (acetic, propionic, butyric, valeric, isovaleric acids) and give values for comparison between groups.
4 182-183 It is necessary to explain with more details the different levels in disease effect.
Is not the disease effect confused with diet effect? Note that T1 y T2 are in healthy group and T3 and T4 are in ERE group.
7 194 The term "cecal samples" is confused. Did authors analyse the cecotropes? or did authors take samples from inside of cecum of animals?. Please, authors must clarify it.
Figure 2. Please, indicate the moment (at 32 day or at 70 days age).
Authors should comment on the advantages or disadvantages in each index (Shannon, Simpson, ACE and Chao1), and why are the results different between them?.
Please, the authors must help the reader to interpret figures 3 and 4. What is the usefully of these figures?
Pag line
8 216 The authors misinterpret the P_value. It is not correct to write “Both the Shannon and Chao1 indices showed significantly higher values (P=0.022 and P=0.020 respectively), but no such difference was seen for ACE and Simpson values.”
Note P_value only shows if there is or not differences between groups, this value does cannot use to discuss if the differences are high or low between groups.
Authors should reason the discrepancies in results between indices.
Please, use fecal or faecal samples throughout all text, but not both terms.
In Tables: In order to compare easier the different within groups, the tables 2, 3 and 4 could include letters (a, b, c or d) as in table 5.
Please, authors must always use the same annotation. For example, table 2 shows standard deviation values after +, but in the rest of tables standard deviation values are within parenthesis.
Pag lines
9 254 delete “treatment”
9 255-257 The ratio Firmicutes on Bacteroidetes can be interesting to compare healthy status between groups, cannot it?
In line 102-104 from M&M, authors indicate that information regarding productivity values which were recorded as weekly weight gain, food consumption (FC) and daily weight gain (DWG). Why were not these data analysed and showed in tables?.
Discussion
Why is it interesting to study the microbiota in both hard and soft feces? instead of only the hard or soft feces?.
The hard and soft feces showed large differences in microbiota; then, which can be more informative about healthy status of animal, the microbiota of hard or soft feces?
Why is not the relationship between the different phyla and the VFAs (acetic, propionic, butyric, valeric, isovaleric acids) discussed?
Line 324, the affirmation “The increase in total VFAs (Table 3) is associated with an increased microbial diversity (Tables 4 and 5) is not supported for the tables.
Conclusions
The conclusions are not supported by the results.
Author Response
Dear Reviewer 1
Thank you for the feedback. We can see the improvements in the manuscript: “Effect of feeding insoluble fiber on the microbiome and metabolites of the cecum and feces of rabbits recovering from epizootic rabbit enteropathy relative to non- infected animals”
In this version, we have done all corrections and suggestions (see file attached). Hopefully, this manuscript has been further improved by the current changes and is now ready for publication.
Best regards
Andrea Margarita Olvera-Ramírez
Reviewer 2 Report
The manuscript addresses an important issue of the possibility of improvement recovery from epizootic rabbit enteropthy by feeding high fiber diet. The study is well designed. However, the manuscript lacks a logical sequence. The authors did not justify the purpose of the volatile gas and the microbiom composition analysis of the samples. There is no information about what gases composition is desired and what proves disbalance. The same is true for the bacteriological composition. The authors should draw specific conclusions about the effect of the diets used - what was it beneficial or not on enetrophaty recovery, volatile gases compositions and microbial community.More specific comments:
Introduction:
1) Some references about beneficial effect of fibre rich diet should be added for example :DOI: 10.1186/s12864-018-5265-x
2) Rationale for voltile gasses and microbiome analyses should be added
Material and methods:
1)How cecum and fecal samples were distinguished? How many animals were there in one cage? it should be described in material and methods
2) Library preparation step is missing, please provide info rmation about the kit used
3) How many samples were sequenced?
Results:
Table 2 Please provide the information which goups were different from each other
Fig.1 Please provide standard deviation on the graph
Moreover, I suggest that discussion should be divided into subchapters describing each finding of the study separately.
Author Response
Dear Reviewer 2
Thank you for the feedback. We can see the improvements in the manuscript: “Effect of feeding insoluble fiber on the microbiome and metabolites of the cecum and feces of rabbits recovering from epizootic rabbit enteropathy relative to non- infected animals”.
In this version, we have done all corrections and suggestions (see file attached). Hopefully, this manuscript has been further improved by the current changes and is now ready for publication.
Best regards
Andrea Margarita Olvera-Ramírez

Round 2
Reviewer 1 Report
Authors have sent us a very neglected version. I am sorry but I cannot review the article titled " Effect of feeding insoluble fiber on the microbiome and metabolites of the cecum and feces of rabbits recovering from epizootic rabbit enteropathy relative to non-infected animals" since the document has many formatting and English errors.
Authors must pay atention and correct all mistakes before resubmitting their document for review.
Author Response
Reviewer 1 raised no fresh specific points this time. Therefore we feel that any comments beyond the previous reply are generic and dealt with in the tracked version of the current resubmission.
Reviewer 2 Report
The authors have improved the manuscript significantly, however the paper contains many errors. It looks like it has not been checked before submission. Most of my comments has been addresed properly, however I still can not find the information about the number of samples for sequencing in material and methods section
Author Response
Reviewer 2
We thank reviewer for the feedback on the resubmission. We have tried to identify errors which were present and these have been changed where found (see trackers for changes). We have also added things such as superscripts to areas of tables where we noticed they were missing.
The reviewer highlights a specific area of the M&M section which needed to be clarified. We think we understand what is being requested. We have tried to clarify what we think this reviewer is requesting by re-wording the issue of how many samples were used (see line 164). Hopefully this is the issue being requested in terms of numbers of samples being sequenced.